https://doi.org/10.1038/s42003-019-0628-7　**OPEN**

# Oribatid mites show that soil food web complexity and close aboveground-belowground linkages emerged in the early Paleozoic

Ina Schaefer [1,2]* & Tancredi Caruso [1]

The early evolution of ecosystems in Palaeozoic soils remains poorly understood because the fossil record is sparse, despite the preservation of soil microarthropods already from the Early Devonian (~410 Mya). The soil food web plays a key role in the functioning of ecosystems and its organisms currently express traits that have evolved over 400 my. Here, we conducted a phylogenetic trait analysis of a major soil animal group (Oribatida) to reveal the deep time story of the soil food web. We conclude that this group, central to the trophic structure of the soil food web, diversified in the early Paleozoic and resulted in functionally complex food webs by the late Devonian. The evolution of body size, form, and an astonishing trophic diversity demonstrates that the soil food web was as structured as current food webs already in the Devonian, facilitating the establishment of higher plants in the late Paleozoic.

[1] School of Biological Sciences and Institute for Global Food Security, Queen's University Belfast, 19 Chlorine Gardens, BT9 5DL Belfast, UK. [2] JFB Institute of Zoology and Anthropology, University of Göttingen, Untere Karspüle 2, 37073 Göttingen, Germany. *email: ischaef@gwdg.de

Terrestrial ecosystems above and below the ground are intricately linked by nutrient exchange between plants and soil-living decomposers[1]. This implies that aboveground-belowground linkages have evolved for more than 400 million years[2–4] with newly emerging niches and traits that have led to the complex and diverse ecological communities we observe today. The major groups of soil organisms are bacteria, fungi, protists, nematodes and a high number of strikingly diverse microarthropods, which have characteristically small body size (<2 mm), high local abundance (up to 400,000 individuals m$^{-2}$) and a seemingly broad and generalist diet[5]. The two most common and abundant microarthropod groups in soil are oribatid mites (Acari) and Collembola (Hexapoda) and both are well represented in the terrestrial fossil record. The Early Devonian Rhynie Chert record (~410–400 mya) contains one Collembola and five acariform mite specimens (Endeostigmata and Trombidiformes)[6,7] and the Middle Devonian site at Gilboa, NY (~379–376 mya), includes eleven oribatid mite specimens, representing four species[8]; six more oribatid mite species of six genera turn up in the Lower Carboniferous site of County Antrim, NI (~336–326 mya)[9]. This diversity of oribatid mite lineages, which originated in the Palaeozoic, suggests that this group of mites was already functionally diverse in early soils, and that these mites have been evolving within the soil-system ever since. However, the early evolution of terrestrial ecosystems, especially below the ground, is not well known. To infer the development of soil food webs over geological times, oribatid mites are particularly interesting as they have an outstanding fossil record among soil-living arthropods, ranging from the Devonian to more recent deposits in Miocene amber. They are also abundant as subfossils, especially from peat deposits, and are recognized as valuable indicators for palaeoenvironmental reconstructions[10,11]. This fossil record documents major changes in body size and form[12], morphological diversity[13,14], local abundance[15–17], and even behavioral and trophic interactions[18,19]. The body plan of early acariform mites appeared suddenly in the fossil record without transitional characters linking them to earlier fossil chelicerates[20], but many fossils show striking resemblance to extant families and species, demonstrating morphological stasis for millions of years[12,20–22]. This stasis suggests the existence of phylogenetic trait conservatism in this group, which is potentially very important. Stasis, in fact, implies that the functional roles of these species in the food web have remained relatively constant for hundreds of millions of years. Today, oribatid mites play a central role in the architecture of the soil food web. Often, these mites are collectively and functionally described as opportunistic generalists using resources from fungi to algae and decaying plant material[5]. However, stable isotope studies of oribatid mites clearly show that they cover a broad range of trophic levels, which actually comprise all trophic strategies in the soil food web, from detritivores at the base to predators and scavengers at the top of the food web[23].

Considering the central role of oribatid mites in the soil food web and their presence in terrestrial soils for >380 my, functional traits in this taxon likely evolved along with the developing aboveground vegetation and soils and the increasing complexity of roots, soil structure, and organic matter. Traits that reflect differences in soil structure are body size and body form, because both determine the spatial niche oribatid mites occupy in the soil-matrix. Further, body size is not only very consistent within, and often distinct among taxonomic groups of oribatid mites, but also strikingly consistent among fossil and extant taxa[12], suggesting strong functional constraints on morphology. The diversity of trophic strategies and thus food web complexity is reflected by the many trophic levels covered by oribatid mites, which seems consistent within (adult)

species[23] although a formal phylogenetic analysis of traits and trophic strategies has never been conducted.

The study of trait evolution in soil organisms is in its infancy for major groups of animals and it is unclear how morphological and trophic diversification of extant oribatid mites map onto their phylogeny. However, we postulate that morphological and trophic trait diversification in the phylogeny of this ancient animal group reflect functional diversity or major changes in Paleozoic soil food webs and thus in aboveground-belowground linkages[24].

With the overarching goal of shedding light on the early phase of the soil food web evolution and establishment of belowground-aboveground linkages, we used functional traits of extant oribatid mites (morphology and stable isotope data) to conduct, to our knowledge, the first phylogenetic trait-based analysis of oribatid mites and tested for phylogenetic signal in habitat and resource-related traits. For this, we reconstructed the most complete molecular phylogenetic tree of oribatid mites currently available. We also dated our phylogenetic analysis using the available fossil record and related major transitions in oribatid mites to major transitions in vegetation over geological time. Important aboveground transitions that should have facilitated morphological trait evolution and diversification were the development from herbaceous vegetation of leafless plants with shallow rhizoid rooting systems to forest ecosystems with deep roots and high productivity during the Devonian. This was followed by the enormous diversification and expansion of forests into continental areas in the Carboniferous[25,26]. Transitions that could have influenced oribatid mite diversification or induced shifts in feeding strategies were floristic changes and reorganizations of plant communities such as those due to climatic shifts within the Carboniferous[26] or the evolution of flowering plants in the Cretaceous[27]. Specifically, we aimed to analyze the evolution of variation in body size, form, and trophic strategies in relation to the increase in habitat complexity and availability of resources over geological time, and to test if major changes match with functional shifts in aboveground vegetation since the occurrence of vascular plants in the Late Silurian 425 mya.

Our results show a strong phylogenetic signal of morphological differentiation, which occurred very early in the evolution of oribatid mites and contrasts with the lack of trophic differentiation in phylogenetic clades. The differentiation into different sizes and forms occurred in parallel with the evolution of biomes above the ground, suggesting the existence of aboveground-belowground linkages since the early Palaeozoic.

## Results

**Phylogenetic analysis**. We used publicly available sequences from 112 oribatid mite species plus six acarine outgroup species to generate the most extensive phylogeny of oribatid mites today. In this study, we concentrated on the phylogenetic relationships among oribatid mites but excluded the taxon Astigmata. This is now recognized as part of Oribatida[28], but strongly differs in ecology and life histories. Astigmata are mostly parasitic, and free-living species occur patchy in moist environments that are high in organic matter, or in ephemeral habitats, such as decaying logs, fungal fruiting bodies, dung, carrion or tree holes, and therefore play a limited role in terrestrial soil food webs[5,29]. In fact, these mites likely evolved as a monophyletic, highly derived offshoot within oribatid mites[28,30,31]. We thus removed this taxon from our analyses to concentrate on soil species. However, we included three Endoestigmata, which is relevant for the phylogenetic resolution among early-derived oribatid mites and for fossil calibrated divergence time estimations[28]. We included three prostigmatid mites as a outgroup taxon to

Acariformes, but ignored non-mite chelicerate taxa as outgroups in this study, because we specifically focused on trait evolution and divergence times of nodes and branches only within oribatid mites. The species used in this analysis represented all major taxonomic groups of higher (=Brachypylina, Circumdehiscentiae) and lower oribatid mites (=Macropylina, including Desmonomata, Mixonomata, Parhyposomata, Enarthronota, Palaeosomata)[32], which cover the taxonomic and phylogenetic diversity of the entire taxon, and our tree topology is robust and consistent with previously published phylogenies[33–35] (Supplementary Table 1, Supplementary Figs. 1–3, see Methods section for details).

**Molecular divergence times and changes in aboveground vegetation.** The available and substantial fossil record also allowed calibrating a robust molecular clock using a GTR model (Yule tree model and exponential priors). After testing several clock parameters and a combination of 19 potential priors from the fossil record, we identified nine informative priors for calibrating the phylogenetic tree (Supplementary Tables 2, 3, Supplementary Fig. 3) and our results suggest that the early evolution of oribatid mites took place in the Palaeozoic, with 25% of all branching events occurring within a time period of 200 million years, from the Devonian to the Late Triassic (Figs. 1 and 2). However, gaps in frequency of branching times occurred and roughly coincided with important floristic changes and reorganizations of plant communities in the Carboniferous[26]. These major changes occurred at the Devonian-Mississippian boundary (359 mya), the Mississippian-Pennsylvanian boundary (318 mya)

and in the Kasimovian (~305 mya). In the Mesozoic, diversification rates increased from the Triassic to the mid Jurassic, with 50% of all branching events occurring during a period of 160 my, from the Late Triassic to Early Eocene (212–52 mya).

**Phylogenetic signal of morphological and trophic traits.** We investigated functional traits using size measurements and molecular data from extant taxa, and trophic strategies of the same taxa were inferred from stable isotope data. Oribatid mites can be divided into several trophic groups[23,36], depending on their trophic role in the soil food web. We numbered trophic levels (TL) from 0 to 3, whereby 0 are specialist lichen feeders, 1 and 2 are primary and secondary decomposers, respectively, and 3 are predators or scavengers (for details on the definition of trophic levels see the Trait section in Methods). The distribution of trophic levels on the phylogeny clearly demonstrated that primary and secondary decomposers (TL = 1 and 2) are the most common functional groups but that lower (specialists, TL = 0) and higher levels (predators/scavengers, TL = 3) are still common and occur in all clades (Fig. 3).

Phylogenetic signal of body size ($K = 0.44$, $P = 0.0001$; $\lambda = 0.78$, $P < 0.0001$) and form (i.e., length/width ratio: $K = 0.72$, $P = 0.001$; $\lambda = 0.94$, $P < 0.0001$) were statistically significant and indicated convergent evolution. In contrast, phylogenetic signal of trophic level was not statistically significant ($K = 0.14$, $P = 0.89$; $\lambda < 0.0001$, $P = 1$), but phylogenetic independent contrast (PIC) showed that species within several clades were positioned at significantly different trophic levels or accessed different resources (Fig. 3, Supplementary Fig. 4). In contrast to

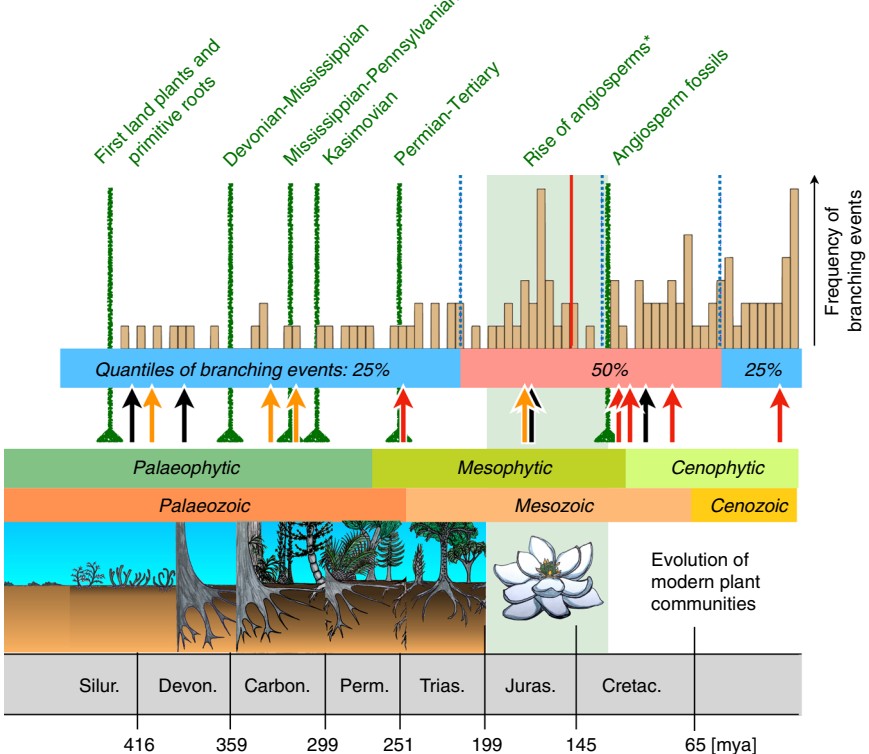

**Fig. 1** Gaps and rises in the branching frequencies correlate with floral changes in the Palaeophytic and Cenophytic. Shifts in body size and body form correlate with the advance of vascular plants in the Devonian and forests in the Carboniferous (black and orange arrows), but most shifts in trophic level (red arrows) occurred in the Cenophytic, parallel to the evolution of angiosperms. The Permian-Tertiary mass extinction is not reflected in the branching frequency, but floral changes such as the extinction of progymnosperms and the diversification of lignophytes at the Devonian-Mississippian boundary and the collapse of rainforest and swamp vegetation in the Kasimovian correlate with gaps in oribatid mite radiations. In contrast, radiations of oribatid mite lineages (50% of branching events) correlate with the advent of angiosperm and the evolution of modern plant communities

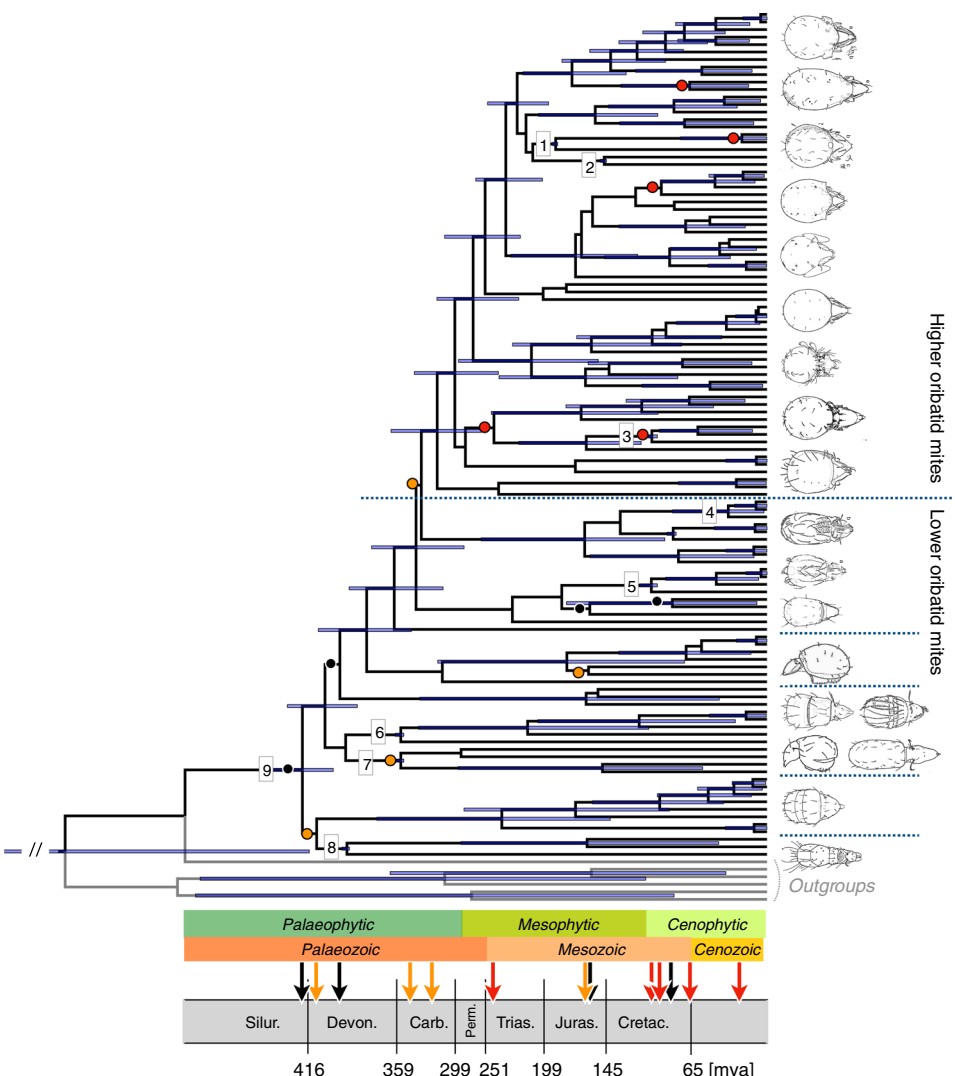

**Fig. 2** Radiation of oribatid mites started early in the Devonian and was accompanied by significant shifts in body size and body form (black and orange arrows on timeline). Lower oribatid mites that evolved in the Devonian likely were small, as body size is phylogenetically conserved in oribatid mites. Different body forms evolved in the Devonian and the Carboniferous into various sizes and in the Triassic. Most shifts in trophic level according to phylogenetic independent contrast (PIC, red arrows) repeatedly occurred since the mid-Cretaceous, with the beginning of the Cenophytic, when angiosperms evolved. Circles on nodes in the phylogeny indicate significant shifts in body size (black), body form (orange) and trophic level (red, see also Fig. 1) and correspond to the arrows on the timeline. Numbers on nodes show the distribution of nodes with taxa known from the fossil record that were used as priors in the molecular clock analysis (1 Hydrozetidae-Limnozetidae, 196–189 mya; 2 Achipteriidae, 145–140 mya; 3 Carabodidae, 122–99 mya; 4 Camisiidae, 85–83 mya; 5 Trhypochthoniidae 122–99 mya; 6–7, Parhyposomata/Enarthronota 336–326 mya; 8 Palaeosomata, 385–374 mya; 9 oribatid mites, 407–385 mya; for details and references see Supplementary information)

morphological changes, trophic shifts only occurred at terminal or internal nodes but never on the backbone of the phylogeny (Fig. 3).

Phylogenetic signal of body size and body form indicated that small and large body sizes, and elongated and round body forms evolved several times in different clades while remaining conserved within some clades (Fig. 3). Early-derived species are dominantly small and elongated but oribatid mite evolution was accompanied by various shifts in these traits during the Devonian and Carboniferous. The round to ovoidal body form was fixed in higher oribatid mites at the end of the Carboniferous. Since the early Triassic, body size and body form remained stable except for one shift in the Jurassic and Cretaceous, each at derived nodes (i.e., within families). Two distinct patterns are thus clear in the evolution of oribatid mites. First, most and major morphological

changes occurred in the Palaeozoic, i.e. significant changes in body size and body form only took place in lower oribatid mite radiations during the Devonian and Carboniferous. Second, all trophic changes, i.e. significant shifts in trophic levels, occurred only in higher oribatid mites and mainly during the Mesozoic, and predominantly during the Cretaceous (Figs. 1 and 2). The dated phylogeny and the inferred distribution of significant shifts among early-derived lineages into various body sizes and body forms indicate that oribatid mites already occupied different spatial niches in early soils. However, the overall lack of phylogenetic signal in trophic level suggests that each of these oribatid mite lineages consumes a variety of resources within their spatial niches, indicating the existence of an already functional food web that covered all trophic levels in the Late Silurian to Early Devonian (~419–385 mya).

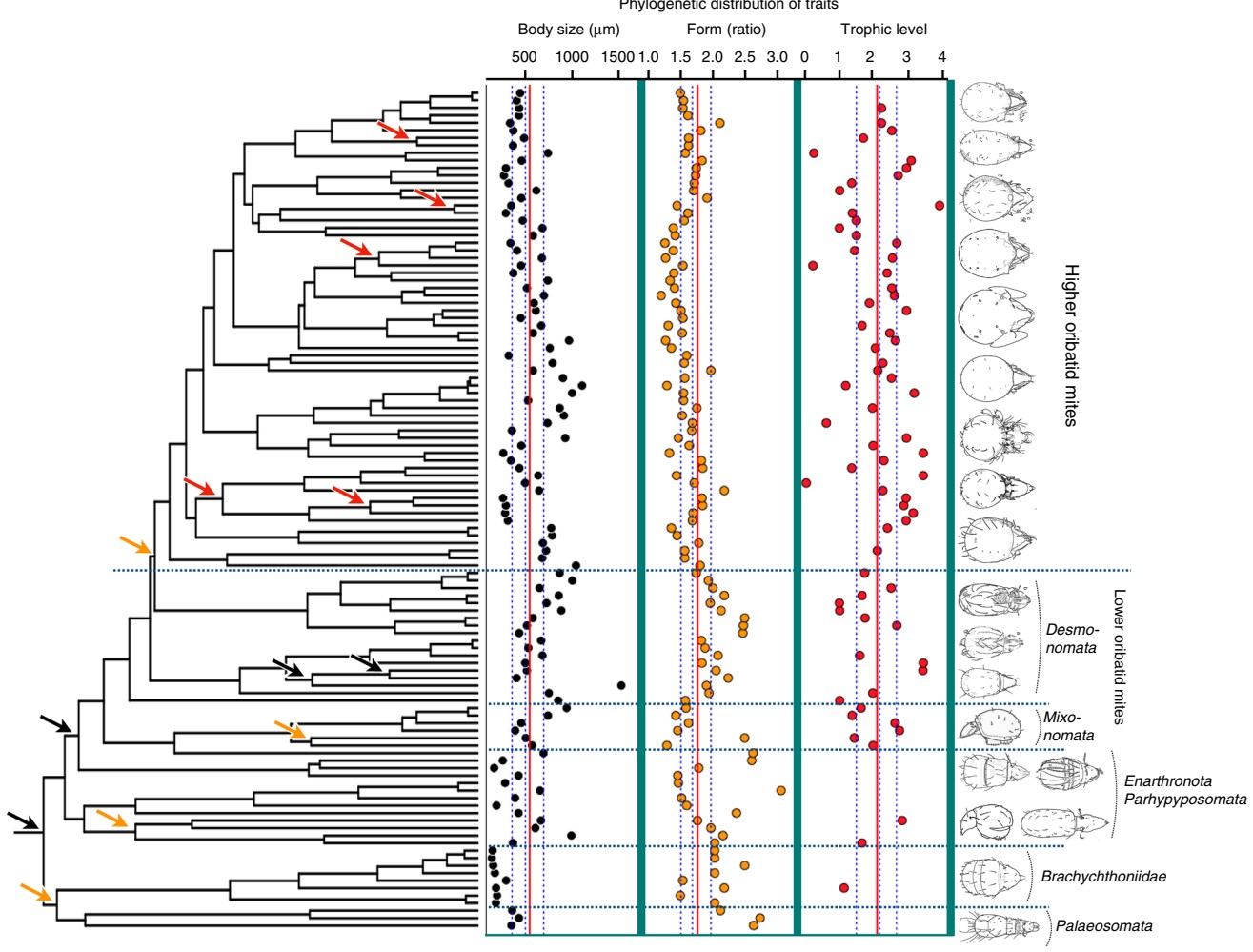

**Fig. 3** Body size and form differs significantly among early-derived oribatid mites and differences are strongly explained by phylogeny. Variation in body size and form is conserved within clades but developed convergent in higher taxa of lower and higher oribatid mites, resulting in significant phylogenetic signal for body size and form. The non-random distribution of data points for body size and form along the mean of the trait distribution (vertical red lines in trait columns) shows that oribatid mites started very small (Palaeosomata and Brachychthoniidae) but various body sizes and forms evolved in other lower oribatid mites (Enarthronota, Parhyposomata, Mixonomata, Desmonomata). The ovoidal, almost round body form became fixed early in higher oribatid mites. In contrast to morphological traits, shifts in trophic level occurred only within higher oribatid mites and showed no phylogenetic signal; all trophic levels occur in all groups. Arrows in the phylogenetic tree indicate nodes with taxa that differed significantly in trait distribution by phylogenetic independent contrast (PIC), i.e. in body size (black arrows, black dots in first trait column), body form (orange arrows, orange dots in second column) and trophic level (red arrows, red dots in third column). Red lines in trait columns show the mean distribution, blue dotted lines the 25% and 75% quantiles and the median of the respective trait. Pictures on the right show examples of body form variation in lower and higher oribatid mites[37]

## Discussion

Small soil animals move in the pore-space of an architecturally complex soil-matrix, and their body size and form determine access to specific segments of a three-dimensional space (e.g., small vs large pore spaces), which provides not only resources but also some enemy-free space. Thus, the current diversity observed in soil animal body size and form could evolve functionally only when the three-dimensional complexity and structure of soil also evolved in terms of aggregates of different forms and sizes, which created pore spaces of different forms and sizes. Our data show that body size and form of initially small and elongated species diversified repeatedly into distinct size-categories in the Devonian and Carboniferous period. The similarity between the phylogenetically conserved body forms and sizes of extant early-derived species and that of Devonian fossils indicate that these body traits, once emerged, have been maintained for hundreds of millions of years within many clades[8,9,12–14,20–22]. The process of diversification of body sizes and forms as inferred by our analysis

also indicates that the occupation of conserved spatial niches (i.e., certain shape and size of soil pores) in soil started in the mid Palaeozoic, a time when also early terrestrial ecosystems above the ground evolved and diversified, parallel to an increasing complexity in root structures[25,38,39]. This process continued for millions of years, corresponding to the increasing complexity of vegetation and soil structure. Additionally, the convergent evolution of body sizes and forms observed in later-derived groups indicates that new spatial niches emerged again over time and were filled by multiple radiations. We thus propose that the multiple radiations of these convergent spatial niches (as defined by body size and form) are reflective of the increasing complexity of the organic layers of soils over time. It is known that, before vascular plants evolved, soils were weakly developed while the accumulation of organic matter by decaying plants is documented from Early Devonian deposits[40,41]. The development of roots and mycorrhiza accelerated the formation of soil, likely since the Late Silurian-Early Devonian[25,42]. Further, the most common and

oldest forest soils (Alfisols and Utisols) that are rich in organic matter have been described already from the Late Devonian and they are common in every subsequent geological period[40]. Thus, the inferred dates of the radiation in body sizes and forms that we have documented in oribatid mites have occurred in parallel with the evolution of the complexity of soil organic layers. Analogous to the evolution of soils, where new kinds of soils add to the existent ones rather than replacing them[39], the evolution of oribatid mites can also be viewed as the addition of new lineages to existing ones, with emerging combinations of body sizes and forms, rather than the replacement of pre-existing lineages.

Our results demonstrate that specific functional traits (body size and body form) are phylogenetically conserved in oribatid mite lineages that diverged in the Palaeozoic (Late Silurian to Carboniferous) and that these traits evolved convergently in lineages since the Mesozoic. The fossil record demonstrates that body sizes of oribatid mites are intriguingly conserved[12], mites started small and remained small throughout their evolutionary history, and we, therefore, argue that oribatid mites occupied distinct spatial niches (pore size and shape) in early soils of the Paleozoic. However, fossils from the Devonian to the Jurassic tended to be larger than contemporary members of the same taxonomic group[12], which correlates with higher atmospheric oxygen levels that also induced gigantism in Paleo- and Mesozoic insects[43]. Nevertheless, accessible space in soil adds to physiological constraints induced by atmospheric oxygen and bigger mites in the past could also indicate that Paleo- and Mesozoic soils had a different, coarser structure than younger soils. These hypotheses will have to be tested in the future, but we show that shifts in body size and body form did not correlate with a shift in resource preferences as inferred by stable isotopes. The implication is that, given a certain size and form fitting a spatial niche in soil, multiple resources in that niche could be exploited. Our analyses show that trophic differentiation among oribatid mite lineages occurred much later than differentiation of size and form, but not sooner than the Mesozoic era. Importantly, not a single family nor genus is restricted to a certain trophic level. According to our results, the general notion of oribatid mites being generalist is to be revised because these animals inhabit a wide range of spatial niches but each species likely covers a specific trophic range in a specific spatial context.

The lack of phylogenetic signal in stable isotope niches of oribatid mites contrasts with the signals found in radiations of aboveground animals. For example, dietary specialization is phylogenetically structured in mammals[44,45] and birds[46]. We hypothesize that access to resources in soil was limited while this access increased over time as soil structural complexity and oribatid body size and form diversified over time (and over temporal scales of tens to hundreds of millions of years). In current soils, the very same resources are accessible at several spatial scales, i.e. from small to large soil pores. Thus, distantly related species can access the same resource but in different size-compartments in the soil-matrix.

Trophic niches are likely phylogenetically conserved in Collembola[47] although a formal, phylogenetically informed trait analysis is missing for this group. Collembola resemble oribatid mites in phylogenetic age[6], ecology[5], and function[48]. However, this resemblance might be only superficial. Differences in habitat and resource utilization might have caused Collembola diversification to proceed through radiation in trophic rather than spatial niches, and the general implication is that oribatid mites and Collembola actually represent two very different functional groups in the soil food web.

The inferred Palaeozoic diversification of oribatid mite lineages, which is also supported by the fossil record[8,9], was discontinuous and stagnation correlated well with floristic changes

in the Carboniferous, a period of multiple climatic changes with alternation of wet and dry periods caused by global cooling. Floral changes imply changes in the organic input from above the ground to the belowground system, due to differences in litter composition and root anatomy in different plant communities. Stagnation in oribatid mite diversification correlates with the extinction of archaeopteridalean, progymnosperms, and the diversification of arborescent lignophytes at the Devonian-Mississippian border, (2) the origination and diversification of cordaiataleans and conifers at the Mississippian-Pennsylvanian boundary, (3) the collapse of rainforest biomes and a shift from lycopsids to tree fern vegetation at the end of the Carboniferous (Kasimovian)[26]. The Permian-Triassic mass extinction is not noticeably reflected in the phylogenetic diversification pattern. The increase of diversification rates in the Jurassic and Cretaceous also correlated well with the period of angiosperm diversification[27]. Taking floral events into account can be difficult because geological boundaries of plant evolution are quite loose and do not necessarily coincide with faunal extinctions[49], but in our case known floral events correlate well with patterns of oribatid mite radiations and suggest a long-term relationship between above- and belowground diversity.

Overall, our data supports the hypothesis that oribatid mites filled all trophic niches within available soil compartments already in Devonian and Carboniferous soils, thereby building up fully functional food webs that could already aid the nutrient cycling and energy fluxes that constitute aboveground-belowground linkages today. The future challenge is to unveil the evolutionary complexity of belowground food webs and how this co-evolved with the aboveground component of terrestrial food webs. Our study shows that exploring the co-evolution and diversification of interactions among soil-living microfauna, mesofauna, microbes, and plants is a key to resolve this challenge.

## Methods

**Phylogenetic analysis**. Data for molecular phylogeny were downloaded from NCBI (www.ncbi.nlm.nih.gov) or provided by collaborators (15 sequences). Only the 18S gene covered all major groups of oribatid mites (Brachypylina, Desmonomata, Mixonomata, Parhyposomata, Enarthronota, Palaeosomata) and provided sufficient phylogenetic resolution. Genome data were available for only four species, so we constructed phylogenies including COI (nucleotide and amino acid sequences) and 28S. However, these two genes reduced the phylogenetic resolution among lower oribatid mites when combined together. Node support and topological resolution decreased when COI and 28S were added to the 18S dataset because these two markers do not contain sufficient phylogenetic information for deep oribatid mite phylogeny due to saturation (COI) or very short fragment size (28S). The molecular dataset comprised 112 oribatid mite species and six acarine outgroup taxa (Supplementary Table 1). Phylogenetic trees were calculated using Maximum Likelihood using the pml function in the package phangorn[50] in R (www.R-project.org) and Bayesian inference in MrBayes v.3.2.6[51] and BEAST v2.4.7[52]. The topologies of our best resolved tree were consistent, with the exception of some taxa that tend to show unstable phylogenetic positions (Supplementary Figs. 1–3).

**Traits**. Oribatid mites live in an architecturally complex soil-matrix, moving in the pore space among soil particles that differ in shape and size and also vary with depth and soil type. Body size (mean, minimum and maximum size) and form (i.e., body ratio = body length/body width) are therefore a good proxy for the key dimensions of the ecological niche of these mites because they determine access to resources and, possibly, predator free space. The body length to body width ratio describes whether mites are elongated (>1) or globular (~1) on a continuous scale. Body length and width were obtained from the literature[37] under the assumption that average literature values, which are typically based on numerous specimens from many locations, are a fair representation of the central tendency of metrics that display variation in space (see also Supplementary Table 4, Supplementary Figs. 4, 5). Stable isotopes ($^{13}$C and $^{15}$N) per se cannot definitively identify actual food sources of species but relative differences in stable isotopes between species and other bits of information (e.g., chelicerae morphology) allow to hypothesise possible food sources and describe the most likely general, long-term diet of animals. Stable isotopes are also useful to discriminate among different carbon sources animals consume and the relative trophic position they occupy[53]. We obtained original $^{13}$C and $^{15}$N data of oribatid mite communities from four habitats in central Europe: deciduous forest soil (Supplementary Table 5), beech forest dead

wood and trunks[54], dry grassland[55], and peatland[56]. For each habitat we calculated mean $^{13}$C and $^{15}$N for all species that were normalized by the baseline provided in the original data. Trophic levels were calculated for each community by ordering mean $^{15}$N values that were then categorized into specialists (lichen feeders, lowest $^{15}$N values), primary and secondary decomposers, and predators or scavengers (highest $^{15}$N values); each trophic group was separated by ~3.4 $\sigma^{15}$N units. The full integer indicates the trophic level, and the decimal numbers the relative position of species within their trophic level. Numbers, therefore, range from 0.0 (lowest $^{15}$N value) to 3.99 (highest $^{15}$N value within the predator/scavenger category). For species with more than one measurement we calculated the mean of all trophic levels. In total, we compiled stable isotope values for 70 oribatid mite species (Supplementary Table 5, Supplementary Fig. 5).

**Molecular divergence time estimation.** We estimated divergence times in BEAST v2.4.7[52] using the GTR site model, the Yule tree model and exponential priors with an mcmc chain of 50 million generations and a sampling frequency of 5000 and a burn-in of 25%. We tested several clock models, clock rates and priors, with the relaxed log normal clock and a rate of 0.02 and 0.0005 performing best. We tested combinations of 19 potential priors from the fossil record, and eliminated those priors that impeded chain convergence and had poor ESS values, resulting in nine informative priors for calibration of the phylogenetic tree (Supplementary Table 2). Chain convergence was checked on Tracer v.1.6[57] and the final phylogenetic tree was generated with TreeAnnotator v.2.4.7 using a burn-in of 25% and a posterior probability limit of 0.8. The distribution of branching times was analyzed with the ips package in R[58], plotting bins of 5 million years intervals. Major aboveground events (green vertical lines in Fig. 1) were selected from the literature[26,27], i.e. the emergence of early vascular plants and primitive roots (425 mya), the Devonian-Mississippian (359 mya), the Mississippian-Pennsylvanian (318 mya) and the Kasimovian (303–307 mya) biome changes, and the first fossil record of angiosperm fossils (124 mya). Significant trait divergences from the pic analyses were added to the branching times by transferring the mean divergence times of the respective nodes from the dated phylogenetic tree to the plot.

**Quantifying phylogenetic signal.** Based on the dated phylogeny generated by BEAST, we quantified the phylogenetic signal of all traits using Blomberg's K[59] and Pagel's λ[60,61] with random 9999 replicates using the R packages phytools[62] and geiger[63]. To quantify the size of trait divergence, i.e. if a trait at a node is larger or smaller than expected, we used the pic function[64] implemented in the ape package[65] using 9999 random replicates. Stable isotope data of several early-derived taxa, which are rare and very small, were not available and trophic traits were thus analyzed with a separate phylogeny to reduce a bias in phylogenetic signal caused by missing data.

**Reporting summary.** Further information on research design is available in the Nature Research Reporting Summary linked to this article.

## Data availability
All datasets generated or analyzed during this study are included in this published article (and its Supplementary Information files) or are available from the corresponding author on request.

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

## Acknowledgements
T.C. and I.S. were supported by the project SENSE (Structure and Ecological Niche in the Soil Environment; EC FP7 - 631399 - SENSE). We thank Stefan Scheu and Mark Maraun for providing 18S sequences of oribatid mite species and commenting on an earlier version of this paper.

## Author contributions
I.S. compiled the data, performed the analysis and prepared the figures. T.C. and I.S. conceived the idea and wrote the paper.

## Competing interests
The authors declare no competing interests.
