## [Peer Review File · Communications Biology]

Reviewers' comments:

Reviewer #1 (Remarks to the Author):

OVERVIEW

This manuscript offers a comprehensive, fossil-calibrated molecular phylogeny as a basis for understanding the diversification of oribatid mites back in deep time and their long-term role in soil ecosystems. Given the abundance of oribatids in many modern soil and litter communities - and their important role here as, e.g., decomposers - this manuscript would certainly be of interest to ecologists, mite workers, soil biologists and even palaeontologists.

Although this manuscript has potential, as shown by the collection of much important data, my main concern is the way in which the results have been presented for an interdisciplinary journal. I realise the authors are probably working to a word limit in the main text, but I felt that in places the results (and their significance) could be better explained, should appear in the manuscript in a different sequence or else assume too much prior knowledge from an wider readership.

For example, in lines 89-92 you mention "important floristic changes", but do not tell us what these changes are and/or why they may have been significant for the mites. In fairness this is to some extent explained later (lines 236-238), but I think it would be better to give this prominence in the manuscript where vegetational changes are first mentioned.

Around lines 102-105 you mention trophic levels. However I'm missing a clear introductory statement here for non-ecologists about what trophic levels are with respect to oribatids, or at least a reference to where this slightly cryptic scheme of "TL 0-3" comes from and/or cross-referencing to the Methods (lines 217-220). Perhaps it would be better to say something in the Results like "Oribatids can be divided into several ecological groups which are conventionally numbered trophic levels (TL) 0-3, whereby 0 are specialist lichen feeders, 1 and 2 are primary and secondary decomposers respectively, and 3 are predators or scavengers". Alternatively, say something like: "for a definition of trophic levels see Methods (Traits)".

I feel I am missing the logical link (particularly around lines 108-112) from the data presented to the conclusion that food webs, size distributions and trophic structure must have been diverse back in the Devonian. I assume you are using the fact that your divergence times in the phylogenetic tree imply that numerous clades should have already been present in the mid-Devonian; although I'm not sure this is explicitly stated in the manuscript.

Likewise you assume that early members of a given lineage were morphologically and ecologically similar to living members of these Groups, but is this assumption supported the known fossil record? In general I think you need to be clearer from the outset about whether things like the evolution of body size, body shape and associated trophic diversity are INFERRED (from your phylogeny) or OBSERVED (from the known fossils)?

Some of this is covered in the Discussion, but I think it would be better to have a statement in the Introduction or Results along the lines of early members of lineages are assumed to have a similar body plans and/or lifestyles to their modern counterparts (perhaps with a couple of fossil examples in support of this), thus a high diversity of lineages originating in the Palaeozoic can be reasonably used to infer a similar diversity of trophic levels; something like that?

In this context you need to argue that there has not been too much change in body size within groups over geological time. I'd particularly urge the authors to consult Katya Sidorchuk's recent study of mite body size in the fossil record (Sidorchuk 2019: *Int J Acarol*):

<<https://www.tandfonline.com/doi/abs/10.1080/01647954.2018.1497085>>

Other specific points are detailed below.

INTRODUCTORY PARAGRAPH

It's probably better to say that arthropods and vascular plants were colonised by the Silurian at the latest. We could potentially find older records, and there are even proposals of (terrestrial) spores back into the Ordovician (Rubenstein et al. 2010. *New Phytologist*).

<<https://nph.onlinelibrary.wiley.com/doi/full/10.1111/j.1469-8137.2010.03433.x>>

Again, perhaps better to say you "infer" (not show) that oribatids diversified in the Palaeozoic. I assume

some/much of this implied diversity is based on calibrated branching events in your molecular phylogeny, rather than the handful of actual Devonian/Carboniferous fossil specimens.

INTRODUCTION

lines 37-40: This statement is misleading. The Early Devonian Rhynie mites are NOT oribatids, but may be endeostigmatids or, perhaps, Tyeidae. The oldest unequivocal oribatids are from Gilboa (Middle Devonian). See, e.g., the review of Rhynie mites in Dunlop & Garwood (2018) Phil Trans R Soc: <<https://www.ncbi.nlm.nih.gov/pubmed/29254958>>

On the other hand, fossil springtails are of course known at Rhynie, but not to my knowledge from Gilboa.

line 48: You could mention that oribatids are also very abundant as subfossils, especially from peat deposits.

RESULTS

Lines 106-117: Are the inferred shifts in body traits based on direct observations of fossils, or are they implied by the dated phylogenetic tree; or do both sets of data support this? Again, do we know the extent to which size and ecology changed within a given group over geological time?

DISCUSSION

lines 144-146: Again, you should probably be clear whether your documented radiation dates in body size and shape are based on the phylogenetic tree or direct observations from fossils.

line 176: Again, what exactly are the major biome changes you refer to here: the end Devonian mass extinction, the rise (and eventual collapse) of the Carboniferous coal forests?

FIGURE 1

This is the primary image associated with the manuscript, but has a number of problems.

Names of major taxonomic groups are not given on the figure (presumably for clarity), but at the same time the terms "higher" and "lower" oribatids are not mentioned or explained in the Introduction or Results. For non-acarologists, these terms are obscure.

You explain the yellow and grey stars, but not the pink/red stars. Are these shifts in trophic level?

There is a trophic level 4 in the figure, but this is not mentioned/defined in the text as far as I can tell.

What are the dashed lines either side of the solid red lines (confidence intervals, standard deviations?)

The red dots for trophic level are in panel b, not c.

Fossil calibrations are not indicated, nor are there any error margins for the dates (I realise this may be hard to fit on the figure without making it more complicated).

Note: "rise of angiosperms" [not raise]

MINOR CORRECTIONS

line 70: "...major transitions..." [not transition]

line 168: "...niches are likely phylogenetically..."

line 178: "Taking floral events into account too..."

line 233: "...using a burn-in of..." [not burnin-in?]

line 249: better "Acknowledgments" ?

line 263: "Gilboa" [not Golboa]

line 269: "A new species of..." ? [remove capitals]

Jason A. Dunlop

Reviewer #2 (Remarks to the Author):

The submitted MS deals with the question of the evolution of food webs over the time. As a model system, the Authors used a soil ecosystem and one of the key groups of soil microarthropods – mites.

Generally, authors discovered that: (1) the complete soil food webs originated already in Devonian, (2) the rate of branching events in oribatids coincided with floral changes, and (3) the size and different body shape evolved several times in some oribatid clades while in others, especially basal ones, were conserved.

This scenario seems to be well supported by obtained data and adequate methodology but I have some reservations that should be cleared before the publication.

Authors seem to ignore two crucial acarological questions:

1. Are mites monophyletic?

2. Are astigmatid mites a sistergroup of oribatids or Astigmata "grow out" of paraphyletic Oribatida?

Both were resolved in some last years by mean of molecular phylogenetics and comprehensive morphological analyses which supported hypotheses that mites are diphyletic and Astigmata is one of the lineages of Oribatida. Both may have important influence on the results obtained by authors. Firstly, why authors use Endeostigmata and Parasitiformes as outgroups? Endeostigmata are most probably polyphyletic and at least a part of them are members of the Trombidiformes. The sistergroup of Sarcotiformes (Oribatida s.l. + some Endeostigmata) are solifugids, not parasitiform mites. The long branches together with absence of calibration points for outgroups may disturb the molecular dating. I get the feeling (Fig. S3) that for these reasons the stem of Oribatida is shifted down to 525 MYA that is unrealistic. Secondly, it is also difficult to state how the passing over Astigmata has affected both obtained results, molecular dating and scenario of the evolution of web food. Astigmata are members of the clade Oribatida (they are the "higher" oribatids) with completely different life strategies and very accelerated substitution rate. Summing up, the Authors are not allowed to define mites they analyze as a clade or a natural taxon.

Minor remark: Fig. 1 is a quite pretty illustration but the "all-in-one" strategy leads to confusion and is more like puzzle than clear results presentation. Besides, there is an error in the figure caption, see:

"panel c, red dots", "panel b, green bars". I suggest to divide the figure into 2, max. 3 separate figures.

To sum up, Authors should check alternative outgroups with fossil records and full set of oribatid lineages (with Astigmata) before possible publishing in COMMSBIO.

Referee #1: Mite phylogenetics

Referee #2: Molecular phylogeny of mites

Thank you very much for the positive and helpful comments and the interest in our manuscript.

We appreciate the comments on our work and have answered each of your points below.

Reviewers' comments:

Reviewer #1 (Remarks to the Author):

OVERVIEW

This manuscript offers a comprehensive, fossil-calibrated molecular phylogeny as a basis for understanding the diversification of oribatid mites back in deep time and their long-term role in soil ecosystems. Given the abundance of oribatids in many modern soil and litter communities - and their important role here as, e.g., decomposers - this manuscript would certainly be of interest to ecologists, mite workers, soil biologists and even palaeontologists.

Although this manuscript has potential, as shown by the collection of much important data, my main concern is the way in which the results have been presented for an interdisciplinary journal. I realise the authors are probably working to a word limit in the main text, but I felt that in places the results (and their significance) could be better explained, should appear in the manuscript in a different sequence or else assume too much prior knowledge from an wider readership.

We appreciate your helpful suggestions and agree that we need to improve clarity and structure.

We incorporated more information in the revised manuscript as outlined in detail below.

1.1 For example, in lines 89-92 you mention "important floristic changes", but do not tell us what these changes are and/or why they may have been significant for the mites. In fairness this is to some extent explained later (lines 236-238), but I think it would be better to give this prominence in the manuscript where vegetational changes are first mentioned.

Reply: Thank you for this advice, we added information in the Introduction and Discussion on the type of ecological changes that likely were relevant for oribatid mite evolution. See Introduction new lines 87-95, where we now explain that: "Important aboveground transitions that should have facilitated morphological trait evolution and diversification were the development from herbaceous vegetation of leafless plants with shallow rhizoid rooting systems to forest ecosystems

with deep roots and high productivity during the Devonian. This was followed by the enormous diversification and expansion of forests into continental areas in the Carboniferous^{25,26}. Transitions that could have contained oribatid mite diversification or induced shifts in feeding strategies were floristic changes and reorganizations of plant communities such as those due to climatic shifts within the Carboniferous²⁶ or the evolution of flowering plants in the Cretaceous²⁷.”

See also our more detailed definition in the Discussion, where we added six new lines (lines 240 - 46): “Floral changes imply changes in the organic input from above the ground to the belowground system, due to differences in litter composition and root anatomy in different plant communities. Stagnation in oribatid mite diversification correlate with the extinction of archaeopteridalean, progymnosperms, and the diversification of arborescent lignophytes at the Devonian-Mississippian border, (2) the origination and diversification of cordaitaleans and conifers at the Mississippian-Pennsylvanian boundary, and (3) the collapse of rainforest biomes and a shift from lycopsids to tree fern vegetation at the end of the Carboniferous (Kasimovian)²⁶.”

1.2 Around lines 102-105 you mention trophic levels. However I'm missing a clear introductory statement here for non-ecologists about what trophic levels are with respect to oribatids, or at least a reference to where this slightly cryptic scheme of "TL 0-3" comes from and/or cross-referencing to the Methods (lines 217-220). Perhaps it would be better to say something in the Results like "Oribatids can be divided into several ecological groups which are conventionally numbered trophic levels (TL) 0-3, whereby 0 are specialist lichen feeders, 1 and 2 are primary and secondary decomposers respectively, and 3 are predators or scavengers". Alternatively, say something like: "for a definition of trophic levels see Methods (Traits)".

Reply: Thank you for this remark; we now describe trophic levels briefly in the Introduction and more clearly in the Results. Specifically, in the Introduction (at lines 62-65) we say “However, stable isotope studies of oribatid mites clearly show that they cover a broad range of trophic levels, which actually comprise all trophic strategies in the soil food web, from detritivores at the base to predators and scavengers at the top of the food web²³.”

In the Results (see lines 135-142) we now say: “Oribatid mites can be divided into several trophic groups^{23,34}, depending on their trophic role in the soil food web. We numbered trophic levels (TL) from 0 to 3, whereby 0 are specialist lichen feeders, 1 and 2 are primary and secondary decomposers, respectively, and 3 are predators or scavengers (for details on the definition of trophic levels see the Trait section in Methods). The distribution of trophic levels on the phylogeny clearly demonstrated that primary and secondary decomposers (TL=1 and 2) are the most common functional groups but that lower (specialists, TL=0) and higher levels (predators/scavengers, TL=3)

are still common and occur in all clades (Fig. 3).”

1.3 I feel I am missing the logical link (particularly around lines 108-112) from the data presented to the conclusion that food webs, size distributions and trophic structure must have been diverse back in the Devonian. I assume you are using the fact that your divergence times in the phylogenetic tree imply that numerous clades should have already been present in the mid Devonian; although I'm not sure this is explicitly stated in the manuscript.

Reply: On reflection, we agree that there is some lack of logical flow in data presentation and interpretation and have used this comments as an opportunity to add more explicit statements in the manuscript as outlined below. In the Results (lines 162-168) we say that “The dated phylogeny and the inferred distribution of significant shifts among early derived lineages into various body sizes and body forms indicate that oribatid mites already occupied different spatial niches in early soils. However, the overall lack of phylogenetic signal in trophic level suggests that each of these oribatid mite lineages consumed a variety of resources within their spatial niches, indicating the existence of an already functional food web that covered all trophic levels in the Late Silurian to Early Devonian (~419-385 mya).”

We also revised the Introduction and added background information that support and explain the logical (i.e. phylogenetic) link between the function of extant and palaeozoic oribatid mite species. Now, in the Introduction (see lines 54-60) we say that “The body plan of early acariform mites appeared suddenly in the fossil record without transitional characters linking them to earlier fossil chelicerates²⁰, but many fossils show striking resemblance to extant families and species, demonstrating morphological stasis for millions of years^{12,20-22}. This stasis suggests the existence of phylogenetic trait conservatism in this group, which is potentially very important. Stasis in fact implies that the functional roles of these species in the food web have remained relatively constant for hundreds of millions of years.” In the Introduction we explain in nine additional lines (66-75) that “Considering the central role of oribatid mites in the soil food web and their presence in terrestrial soils for >380 my, functional traits in this taxon likely evolved along with the developing aboveground vegetation and soils and the increasing complexity of roots, soil structure and organic matter. Traits that reflect differences in soil structure are body size and body form, because both determine the spatial niche oribatid mites occupy in the soil matrix. Further, body size is not only very consistent within, and often distinct among taxonomic groups of oribatid mites, and also strikingly consistent among fossil and extant taxa¹², suggesting strong functional constraints on morphology. The diversity of trophic strategies and thus food web complexity is reflected by the many trophic levels covered by oribatid mites, which is consistent within (adult) species²³ although

a formal phylogenetic analysis of traits and trophic strategies has never been conducted.”

We also revised and moved down statements from an earlier paragraph in the Introduction (former lines 59-64) to lines 76-80 and added a more explicit statement that we inferred ancient diversity from the phylogeny in the revised paragraph. We now say that “The study of trait evolution in soil organisms is in its infancy for major groups of animals and it is unclear how morphological and trophic diversification of extant oribatid mites maps onto their phylogeny. However, we postulate that morphological and trophic trait diversification in the phylogeny of this ancient animal group reflects functional diversity or major changes in Paleozoic soil food webs and thus in aboveground-belowground linkages²⁴.”

1.4 Likewise you assume that early members of a given lineage were morphologically and ecologically similar to living members of these Groups, but is this assumption supported the known fossil record? In general I think you need to be clearer from the outset about whether things like the evolution of body size, body shape and associated trophic diversity are INFERRED (from your phylogeny) or OBSERVED (from the known fossils)?

Some of this is covered in the Discussion, but I think it would be better to have a statement in the Introduction or Results along the lines of early members of lineages are assumed to have a similar body plans and/or lifestyles to their modern counterparts (perhaps with a couple of fossil examples in support of this), thus a high diversity of lineages originating in the Palaeozoic can be reasonably used to infer a similar diversity of trophic levels; something like that?

Reply: Thank you for this useful comment, we added a statement on Palaeozoic oribatid mite diversity in the Introduction at lines 40-47 where we now say that “The Early Devonian Rhynie Chert record (~410-400 mya) contains one Collembola and five acariform mite specimens (Endeostigmata and Trombidiformes)^{6,7} and the Middle Devonian site at Gilboa, NY (~379-376 mya), includes eleven oribatid mite specimens, representing four species⁸; six more oribatid mite species of six genera turn up in the Lower Carboniferous site of County Antrim, NI (~336-326 mya)⁹. This diversity of oribatid mite lineages, which originated in the Palaeozoic, suggests that this group of mites was already functionally diverse in early soils, and that these mites have been evolving within the soil-system ever since.” Also, at lines 48-51 in the Introduction we state that “To infer the development of soil food webs over geological times, oribatid mites are particularly interesting as they have an outstanding fossil record among soil-living arthropods, ranging from the Devonian to more recent deposits in Miocene amber.”

In the Discussion (new lines 204-212) we clarify that “The fossil record demonstrates that body sizes of oribatid mites are intriguingly conserved¹², mites started small and remained small

throughout their evolutionary history, and we therefore argue that oribatid mites occupied distinct spatial niches (pore size and shape) in early soils of the Paleozoic. However, fossils from the Devonian to the Jurassic tended to be larger than contemporary members of the same taxonomic group¹², which correlates with higher atmospheric oxygen levels that also induced gigantism in Paleo- and Mesozoic insects⁴⁰. Nevertheless, accessible space in soil adds to physiological constraints induced by atmospheric oxygen and bigger mites in the past could also indicate that Paleo- and Mesozoic soils had a different, coarser structure than younger soils.”

We also added a statement on the relevance of fossil data for our results later in the Discussion (lines 237-239) saying that “The inferred Palaeozoic diversification of oribatid mite lineages, which is also supported by the fossil record^{8,9}, was discontinuous and stagnation correlated well with floristic changes in the Carboniferous, a period of climatic changes with wetter and dryer periods caused by global cooling.”

Reply: Additionally, we added statements in two following paragraphs the Introduction lines (76-84) to be more clear about the data we used and to be clear on when we used inferred or observed data (see also our reply to comment 1.3) “The study of trait evolution in soil organisms is in its infancy for major groups of animals and it is unclear how morphological and trophic diversification of extant oribatid mites maps onto their phylogeny. However, we postulate that morphological and trophic trait diversification in the phylogeny of this ancient animal group reflects functional diversity or major changes in Paleozoic soil food webs and thus in aboveground-belowground linkages²⁴.

With the overarching goal of shedding light on the early phase of the soil food web evolution and establishment of belowground-aboveground linkages, we used functional traits of extant oribatid mites (morphology and stable isotope data) to conduct the first phylogenetic trait-based analysis of oribatid mites and tested for phylogenetic signal in habitat and resource related traits.”

More explicit statements on inferences and observations are also added in the following sentences in Results (lines 134-135) “We investigated functional traits using size measurements and molecular data from extant taxa, and trophic strategies of the same taxa were inferred from stable isotope data.” And also in Results (new lines 162-168) “The dated phylogeny and the inferred distribution of significant shifts among early derived lineages into various body sizes and body forms indicate that oribatid mites already occupied different spatial niches in early soils. However, the overall lack of phylogenetic signal in trophic level suggests that each of these oribatid mite lineages consumed a variety of resources within their spatial niches, indicating the existence of an already functional food web that covered all trophic levels in the Late Silurian to Early Devonian

(~419-385 mya). “

Further, we changed the following sentences to be more explicit on observations and inferences in the Introduction (lines 81-84) “With the overarching goal of shedding light on the early phase of the soil food web evolution and establishment of belowground-aboveground linkages, we used functional traits of extant oribatid mites (morphology and stable isotope data) to conduct the first phylogenetic trait-based analysis of oribatid mites and tested for phylogenetic signal in habitat and resource related traits.”

And also in the Discussion (lines 181-185) where we now say that “The process of diversification of body sizes and forms as inferred by our analysis also indicates that the occupation of conserved spatial niches (i.e. certain shape and size of soil pores) in soil started in the mid Palaeozoic, a time when also early terrestrial ecosystems above the ground evolved and diversified, parallel to an increasing complexity in root structures^{25,35,36}.”

Discussion (lines 196-198) “Thus, the inferred dates of the radiation in body sizes and forms that we have documented in oribatid mites have occurred in parallel with the evolution of the complexity of soil organic layers.”

Discussion (lines 247-249) “The inferred increase of diversification rates in the Jurassic and Cretaceous also correlated well with the period of angiosperm diversification²⁷.”

Discussion (lines 249-252) “Taking floral events into account can be difficult because geological boundaries of plant evolution are quite loose and do not necessarily coincide with faunal extinctions⁴⁶, but in our case known floral events correlated well with patterns of oribatid mite radiations and suggest a long-term relationship between above- and belowground diversity.”

1.5 In this context you need to argue that there has not been too much change in body size within groups over geological time. I'd particularly urge the authors to consult Katya Sidorchuk's recent study of mite body size in the fossil record (Sidorchuk 2019: Int J Acarol):

<https://www.tandfonline.com/doi/abs/10.1080/01647954.2018.1497085>

Reply: Thank you for pointing out the missing clarity about inferences and observations in our manuscript. For analyses we only used data of recent taxa, but we included the publication of Sidorchuk 2018 into the manuscript (Reference No. 12) in the Introduction and Discussion as outlined below.

Introduction (lines 52-54) “This fossil record documents major changes in body size and shape¹², morphological diversity^{13,14}, local abundance¹⁵⁻¹⁷, and even behavioral and trophic interactions^{18,19}.”

Introduction (new lines 70-73) “Further, body size is not only very consistent within, and often distinct among taxonomic groups of oribatid mites, but also strikingly consistent among fossil and extant taxa¹², suggesting strong functional constraints on morphology.”

We also added the reference in the Discussion (lines 178-181) “The similarity between the phylogenetically conserved body shapes and sizes of extant early-derived species and that of Devonian fossils indicate that these body traits, once emerged, have been maintained for hundreds of millions of years within many clades^{8,9,12-14,20-22}.”

We also included more detailed information on body sizes of fossil oribatid mites in the Discussion (new lines 204-214) “The fossil record demonstrates that body sizes of oribatid mites are intriguingly conserved¹², mites started small and remained small throughout their evolutionary history, and we therefore argue that oribatid mites occupied distinct spatial niches (pore size and shape) in early soils of the Paleozoic. However, fossils from the Devonian to the Jurassic tended to be larger than contemporary members of the same taxonomic group¹², which correlates with higher atmospheric oxygen levels that also induced gigantism in Paleo- and Mesozoic insects⁴⁰. Nevertheless, accessible space in soil adds to physiological constraints induced by atmospheric oxygen and bigger mites in the past could also indicate that Paleo- and Mesozoic soils had a different, coarser structure than younger soils. These hypotheses will have to be tested in the future, but we show that shifts in body size and body form did not correlate with a shift in resource preferences as inferred by stable isotopes.”

Other specific points are detailed below.

INTRODUCTORY PARAGRAPH

1.6 It's probably better to say that arthropods and vascular plants were colonised by the Silurian at the latest. We could potentially find older records, and there are even proposals of (terrestrial) spores back into the Ordovician (Rubenstein et al. 2010. New Phytologist).

<https://nph.onlinelibrary.wiley.com/doi/full/10.1111/j.1469-8137.2010.03433.x>

Reply: We changed the statement in the Abstract (lines 20-21), now saying that “Early Paleozoic soils were first colonized by arthropods and vascular plants by the Silurian.” and added this reference (reference number 2), and two more references (reference numbers 4 and 5), in the second sentence of the Introduction (line 33-35), where we say that “This implies that

aboveground-belowground linkages have evolved for more than 400 million years^{2,3,4} with newly emerging niches and traits that have led to the complex and diverse ecological communities we observe today.”

The newly added references are:

2. Rubinstein, C. V., Gerrienne, P., de la Puente, G. S., Astini, R. A. & Steemans, P. Early Middle Ordovician evidence for land plants in Argentina (eastern Gondwana). *New Phytol.* **188**, 365-369 (2010).
3. Hagström, J. & Mehlqvist, K. The dawn of terrestrial ecosystems on Baltica: first report on land plant remains and arthropod coprolites from the Upper Silurian of Gotland, Sweden. *Palaeogeogr. Palaeoclimatol.* **317-318**, 162-170 (2012).
4. Kenrick, P., Wellmann, C. H., Schneider, H. & Edgecombe, G. D. A timeline for terrestrialization: consequences for the carbon cycle in the Palaeozoic. *Phil. Trans. R. Soc. B* **367**, 519-536 (2012).

1.7 Again, perhaps better to say you "infer" (not show) that oribatids diversified in the Palaeozoic. I assume some/much of this implied diversity is based on calibrated branching events in your molecular phylogeny, rather than the handful of actual Devonian/Carboniferous fossil specimens.

Reply: Thank you for this important advice and we clarified the statements on inferred diversifications in the Abstract (lines 26-28) where we say that “We conclude that this group, central to the trophic structure of the soil food web, richly diversified in the early Paleozoic, in a process that resulted in functionally complex food webs by the late Devonian.”

We also changed our statements to “inferred” in the relevant sentences in the Discussion (lines 181-185) where we now say that “The process of diversification of body sizes and forms as inferred by our analysis also indicates that the occupation of conserved spatial niches (i.e. certain shape and size of soil pores) in soil started in the mid Palaeozoic, a time when also early terrestrial ecosystems above the ground evolved and diversified, parallel to an increasing complexity in root structures^{25,35,36}.”

And also in the Discussion (new lines 237-239) where we say that “The inferred Palaeozoic diversification of oribatid mite lineages, which is also supported by the fossil record^{8,9}, was discontinuous and stagnation correlated well with floristic changes in the Carboniferous, a period of climatic changes with wetter and dryer periods caused by global cooling.”

And also in the Discussion in lines 247-249 where we now say that “The inferred increase of

diversification rates in the Jurassic and Cretaceous also correlated well with the period of angiosperm diversification²⁷.”

INTRODUCTION

1.8 lines 37-40: This statement is misleading. The Early Devonian Rhynie mites are NOT oribatids, but may be endeostigmatids or, perhaps, Tyeidae. The oldest unequivocal oribatids are from Gilboa (Middle Devonian). See, e.g., the review of Rhynie mites in Dunlop & Garwood (2018) Phil Trans R Soc: <https://www.ncbi.nlm.nih.gov/pubmed/29254958>

On the other hand, fossil springtails are of course known at Rhynie, but not to my knowledge from Gilboa.

Reply: Thank you very much for this remark! We included information on the taxa found in each of the two fossil sites in the Introduction (new lines 40-45) where we now say for greater clarity and accuracy. “The Early Devonian Rhynie Chert record (~410-400 mya) contains one Collembola and five acariform mite specimens (Endeostigmata and Trombidiformes)^{6,7} and the Middle Devonian site at Gilboa, NY (~379-376 mya), includes eleven oribatid mite specimens, representing four species⁸; six more oribatid mite species of six genera turn up in the Lower Carboniferous site of County Antrim, NI (~336-326 mya)⁹.”

1.9 line 48: You could mention that oribatids are also very abundant as subfossils, especially from peat deposits.

Reply: Thank you for this suggestion, we added the information on subfossils in the Introduction (new lines 51-52), mentioning that “They are also abundant as subfossils, especially from peat deposits, and are recognized as valuable indicators for palaeoenvironmental reconstructions^{10,11}.”

RESULTS

1.10 Lines 106-117: Are the inferred shifts in body traits based on direct observations of fossils, or are they implied by the dated phylogenetic tree; or do both sets of data support this? Again, do we know the extent to which size and ecology changed within a given group over geological time?

Reply: Thank you for stressing that the distinction of fossil and present day data is not clear cut in our manuscript. In our analyses we exclusively used data of present day oribatid mite taxa and no fossil data. We referred to the fossil record for two reasons: (1) to explain the presumption that

today's oribatid mites likely occupied similar ecological and functional niches because fossil and recent oribatid mites show strong resemblance in morphological traits (e.g. body size) and (2) to date the molecular phylogeny. We added the following statement on the data we used in the Introduction (new lines 81-85) With the overarching goal of shedding light on the early phase of the soil food web evolution and establishment of belowground-aboveground linkages, we used functional traits of extant oribatid mites (morphology and stable isotope data) to conduct the first phylogenetic trait-based analysis of oribatid mites and tested for phylogenetic signal in habitat and resource related traits. For this, we reconstructed the most complete molecular phylogenetic tree of oribatid mites currently available. “

And in the Results (lines 134-135) “We investigated functional traits using size measurements and molecular data from extant taxa, and trophic strategies of the same taxa were inferred from stable isotope data.”

On information about present day and fossil body sizes, please refer to our reply to comment 1.5.

DISCUSSION

1.11 lines 144-146: Again, you should probably be clear whether your documented radiation dates in body size and shape are based on the phylogenetic tree or direct observations from fossils.

Reply: We agree with the Reviewer and clarified our statements in several sentences throughout the text, please see our reply to comment 1.7 where we outlined the respective sentences.

1.12 line 176: Again, what exactly are the major biome changes you refer to here: the end Devonian mass extinction, the rise (and eventual collapse) of the Carboniferous coal forests?

Reply: We agree and added more information on biome changes to the revised manuscript in the Please see our reply to comment 1.1.

We also added more information in 15 new lines in the Discussion from line 237 to 252 where we describe that “The inferred Palaeozoic diversification of oribatid mite lineages, which is also supported by the fossil record^{8,9}, was discontinuous and stagnation correlated well with floristic changes in the Carboniferous, a period of climatic changes with wetter and dryer periods caused by global cooling. Floral changes imply changes in the organic input from above the ground to the belowground system, due to differences in litter composition and root anatomy in different plant communities. Stagnation in oribatid mite diversification correlate with the extinction of archaeopteridalean, progymnosperms, and the diversification of arborescent lignophytes at the

Devonian-Mississippian border, (2) the origination and diversification of cordaitaleans and conifers at the Mississippian-Pennsylvanian boundary, and (3) the collapse of rainforest biomes and a shift from lycopsids to tree fern vegetation at the end of the Carboniferous (Kasimovian)²⁶. The Permian-Triassic mass extinction is not noticeably reflected in the phylogenetic diversification pattern. The inferred increase of diversification rates in the Jurassic and Cretaceous also correlated well with the period of angiosperm diversification²⁷. Taking floral events into account can be difficult because geological boundaries of plant evolution are quite loose and do not necessarily coincide with faunal extinctions⁴⁶, but in our case known floral events correlated well with patterns of oribatid mite radiations and suggest a long-term relationship between above- and belowground diversity.“

More information on the biome changes were also included in the Introduction as outlined in our reply to comment 1.1.

1.13 FIGURE 1

This is the primary image associated with the manuscript, but has a number of problems.

Names of major taxonomic groups are not given on the figure (presumably for clarity), but at the same time the terms "higher" and "lower" oribatids are not mentioned or explained in the Introduction or Results. For non-acarologists, these terms are obscure.

Reply: This is an important note, thank you! We added a brief definition in the Results in lines 115-121, where we say that "The species used in this analysis represented all major taxonomic groups of higher (=Brachypylina, Circumdehiscentiae) and lower oribatid mites (=Macropylina, including Desmonomata, Mixonomata, Parhyposomata, Enarthronota, Palaeosomata), which cover the taxonomic and phylogenetic diversity of the entire taxon, and our tree topology is robust and consistent with previously published phylogenies³¹⁻³³ (Supplementary Table S1, Figs. S1-S3, see Methods section for details)."

We also split Figure 1 into three separate Figures, where we now provide the names of the major groups of oribatid mites.

1.14 You explain the yellow and grey stars, but not the pink/red stars. Are these shifts in trophic level?

Reply: Thank you for noticing, we split Figure 1 into three separate figures and revised the legends

as following:

Figure 1 Body size and form differs significantly among early derived oribatid mites and differences are strongly explained by phylogeny. Variation in body size and form is conserved within clades but developed convergent in higher taxa of lower and higher oribatid mites, resulting in significant phylogenetic signal for body size and form. The non-random distribution of data points for body size and form along the mean of the trait distribution (vertical red lines in trait columns) shows that oribatid mites started very small (Palaeosomata and Brachychthoniidae) but various body sizes and forms evolved in other lower oribatid mites (Enarthronota, Parhyposomata, Mixonomata, Desmonomata). The ovoidal, almost round body form became fixed early in higher oribatid mites. In contrast to morphological traits, shifts in trophic level occurred only within higher oribatid mites and showed no phylogenetic signal; all trophic levels occur in all groups.

Arrows in the phylogenetic tree indicate nodes with taxa that differed significantly in trait distribution by phylogenetic independent contrast (PIC), i.e. in body size (black arrows, black dots in first trait column), body form (orange arrows, orange dots in second column) and trophic level (red arrows, red dots in third column). Red lines in trait columns show the mean distribution, blue dotted lines the 25% and 75% quantiles and the median of the respective trait. Pictures on the right show examples of body form variation in lower and higher oribatid mites.

Figure 2 Radiation of oribatid mites started early in the Devonian and was accompanied by significant shifts in body size and body form (black and orange arrows on timeline). Lower oribatid mites that evolved in the Devonian likely were small, as body size is phylogenetically conserved in oribatid mites. Different body shapes evolved in the Devonian and the Carboniferous into various sizes and in the Triassic. Most shifts in trophic level according to phylogenetic independent contrast (PIC, red arrows) repeatedly occurred since the mid-Cretaceous, with the beginning of the Cenophytic, when Angiosperms evolved.

Circles on nodes in the phylogeny indicate significant shifts in body size (black), body form (orange) and trophic level (red, see also Fig. 1) and correspond to the arrows on the time line. Numbers on nodes show the distribution of nodes with taxa known from the fossil record that were used as priors in the molecular clock analysis (1 Hydrozetidae-Limnozetae, 196-189 mya; 2 Achipteriidae, 145-140 mya; 3 Carabodidae, 122-99 mya; 4 Camisiidae, 85-83 mya; 5 Trhypochthoniidae 122-99 mya; 6-7, Parhyposomata/Enarthronota 336-326 mya; 8 Palaeosomata, 385-374 mya; 9 oribatid mites, 407-385 mya; for details and reference see supplementary informations)

Figure 3 Gaps and rises in the branching frequencies correlate with floral changes in the Palaeophytic and Cenophytic. Shifts in body size and body form correlate with the advance of

vascular plants in the Devonian and forests in the Carboniferous (black and orange arrows), but most shifts in trophic level (red arrows) occurred in the Cenophytic, parallel to the evolution of Angiosperms. The Permian-Tertiary mass extinction is not reflected in the branching frequency, but floral changes such as the extinction of progymnosperms and the diversification of lignophytes at the Devonian-Missippian boundary and the collapse of rainforest and swamp vegetation in the Kasimovian correlate with gaps in oribatid mite radiations. In contrast, radiations of oribatid mite lineages (50% of branching events) correlate with the advent of Angiosperm and the evolution of modern plant communities.

There is a trophic level 4 in the figure, but this is not mentioned/defined in the text as far as I can tell.

Reply: This is actually Trophic Level 3.99, which is the highest category within Trophic Level 3, and is mentioned in the Methods section (lines 291-293) where we say that “The full integer indicates the trophic level, and the decimal numbers the relative position of species within their trophic level. Numbers therefore range from 0.0 (lowest ¹⁵N value) to 3.99 (highest ¹⁵N value within the predator/scavenger category).

For better clarity we improved the quality of this trait column.

What are the dashed lines either side of the solid red lines (confidence intervals, standard deviations?)

Reply: They blue dashed lines represent the 25% and 75% quantiles and the mean, as explained in at the end of the Figure legend, which is outlined in our reply to comment 1.14.

The red dots for trophic level are in panel b, not c.

Reply: The Figure legend has been revised accordingly.

Fossil calibrations are not indicated, nor are there any error margins for the dates (I realise this may be hard to fit on the figure without making it more complicated).

Reply: We agree that information on the molecular clock analysis is missing in Fig. 1. Following this comment and the suggestion of Reviewer 2 we split the main figure in two figures, one representing the phylogeny with trait shifts and trait distribution, the other displaying the dated molecular phylogeny. The legend of Figure 2 now includes the fossil calibrations we used.

Note: "rise of angiosperms" [not raise]

Reply: Thank you for noticing! Changed!

MINOR CORRECTIONS

Reply: We checked all the below mentioned mistakes, but due to a major revision of the main text, some of these mistakes resolved into new text fragments. However, we checked all the below mentioned mistakes and are grateful that the Reviewer took the time and effort to point these out.

line 70: "...major transitions..." [not transition]

Reply: done!

line 168: "...niches are likely phylogenetically..."

Reply: done!

line 178: "Taking floral events into account too..."

Reply: done!

line 233: "...using a burn-in of..." [not burnin-in?]

Reply: done!

line 249: better "Acknowledgments" ?

Reply: done!

line 263: "Gilboa" [not Golboa]

Reply: done!

line 269: "A new species of..." ? [remove capitals]

Reply: done!

Jason A.Dunlop

Reviewer #2 (Remarks to the Author):

The submitted MS deals with the question of the evolution of food webs over the time. As a model system, the Authors used a soil ecosystem and one of the key groups of soil microarthropods –

moos mites. Generally, authors discovered that:(1) the complete soil food webs originated already in Devonian, (2) the rate of branching events in oribatids coincided with floral changes, and (3) the size and different body shape evolved several times in some oribatid clades while in others, especially basal ones, were conserved.

This scenario seems to be well supported by obtained data and adequate methodology but I have some reservations that should be cleared before the publication.

Reply: Thank you very much for reviewing our manuscript and pointing out important issues on the molecular phylogeny and outgroups. We answer the comments point-by-point below, changes are indicated by quotation marks.

Authors seem to ignore two crucial acarological questions:

1. Are mites monophyletic?
2. Are astigmatid mites a sistergroup of oribatids or Astigmata “grow out” of paraphyletic Oribatida?

Both were resolved in some last years by mean of molecular phylogenetics and comprehensive morphological analyses which supported hypotheses that mites are diphyletic and Astigmata is one of the lineages of Oribatida. Both may have important influence on the results obtained by authors.

2.1 Firstly, why authors use Endeostigmata and Parasitiformes as outgroups? Endeostigmata are most probably polyphyletic and at least a part of them are members of the Trombidiformes. The sistergroup of Sarcoptiformes (Oribatida s.l. + some Endeostigmata) are solifugids, not parasitiform mites.

Reply: Thank you for pointing out that we did not explain the use of outgroups in this phylogeny properly. It is indeed an important issue and we added information on this in the Results where we now say in the new lines 109-115 “Outgroups included Prostigmata and Endeostigmata, both are sister groups to oribatid mites and including Endeostigmata is relevant for the phylogenetic resolution among early derived oribatid mites and for fossil calibrated divergence time estimations²⁸. We also included one parasitiform mite to accurately separate Endeostigmata, which are polyphyletic and belong only in part to Acariformes, but also contain lineages belonging to Parasitiformes. In this study, we ignored non-mite chelicerate taxa as outgroups because we specifically focused on trait evolution and divergence times of nodes and branches only within oribatid mites.”

2.2 The long branches together with absence of calibration points for outgroups may disturb the

molecular dating. I get the feeling (Fig. S3) that for these reasons the stem of Oribatida is shifted down to 525 MYA that is unrealistic.

Reply: This is a very reasonable concern about outgroups and their role for molecular clock analyses. Given the importance of this methodological reflection we have acknowledged this point in the Supplementary Information (Supplementary Figures: Phylogeny of oribatid mites based on 18S rDNA) where we added as statement on how this potentially problem might affect our results, given our goals. “The choice of outgroups and their effects on molecular clock estimates: The topology is robust and the mean molecular divergence time estimate of ~415 mya (ranging from 371-447 mya) between [Palaeosomata + Brachychthoniidae] and [Enarthronota + remaining oribatid mites] is reasonable and predating the oldest oribatid mite fossils (i.e., Enarthronota) only by about 25 my. The phylogeny among basal Acariformes and of Acari among Chelicerata is still not resolved with the molecular data currently available, irrespective of the number of genes and the methods used (e.g. secondary structures of rDNA to improve alignment quality) to improve the phylogeny on the base of Acariformes/Acari (e.g. Pepato & Klimov 2015). This study concentrated on nodes within the group of oribatid mites, and used only Acari as outgroups to infer the basal relationships and the origin of the stem group of oribatid mites. The position of Acariformes among Chelicerates is out of the scope of this study and adding more outgroups would not change the internal topology and age of internal nodes of the investigated ingroup (i.e. oribatid mites without Astigmata).”

2.3 Secondly, it is also difficult to state how the passing over Astigmata has affected both obtained results, molecular dating and scenario of the evolution of web food. Astigmata are members of the clade Oribatida (they are the “higher” oribatids) with completely different life strategies and very accelerated substitution rate. Summing up, the Authors are not allowed to define mites they analyze as a clade or a natural taxon.

Reply: Thank you very much for noticing that we need to provide more information on the taxon sampling in our manuscript. We excluded Astigmata because this study focuses only on functional trait evolution of oribatid mites, because they are fundamental in the soil food web and to the aboveground-belowground nutrient linkage. Astigmata in contrast are morphological and functional highly specialised compared to oribatid mites, most are parasitic on warm-blooded animals. Free-living astigmatid mites do occur in soils and can reach high densities in moist and disturbed habitats. However, their overall presence in soils is low and they are not functionally important to the soil food web. Luckily, they are monophyletic and can be easily excluded without affecting the overall phylogeny of oribatid mites, regardless of whether they are sister to or derived within oribatid mites. We would like to point out that we always used the term oribatid

mites (not Oribatida) which is not a phylogenetic term but a term which is rather used in ecology or taxonomy and groups all species with morphological characters of oribatid mites, irrespective of their phylogenetic associations. We added a small paragraph on Astigmata in the beginning of the Results in the new lines 103-109 where we explain that “In this study we concentrated on the phylogenetic relationships among oribatid mites but excluded the taxon Astigmata. This is now recognized as part of Oribatida²⁸, but strongly differs in ecology and life histories, being mostly parasitic and generally absent or not common in soils. Astigmata play a very limited role in terrestrial soil food webs⁵. In fact, these mites likely evolved as a monophyletic, highly derived offshoot within oribatid mites²⁸⁻³¹. We thus removed this taxon from our analyses to concentrate on soil species.”

2.4 Minor remark: Fig. 1 is a quite pretty illustration but the "all-in-one" strategy leads to confusion and is more like puzzle than clear results presentation. Besides, there is an error in the figure caption, see: "panel c, red dots", "panel b, green bars". I suggest to divide the figure into 2, max. 3 separate figures.

Reply: We agree with the reviewer that Figure 1 is too complex. We split this figure into three, one displaying the phylogeny with trait shifts and the trait panels, a second figure shows the molecular clock tree with error bars and trait shifts plus the time-line below, and a third including the branching frequencies along a time-line, to which trait shifts were added. The figure legends have been revised accordingly (please refer to our reply to comment 1.14 of reviewer 1).

2.5 To sum up, Authors should check alternative outgroups with fossil records and full set of oribatid lineages (with Astigmata) before possible publishing in COMMSBIO.

Reply: Please, see our extended replies to these two points in the previous responses. To sum up our replies, we have incorporated a discussion of these limits of our study in the manuscript and, while we certainly agree that a complete phylogeny of Acariformes is essential to resolve mite systematics, we only investigated a specific, functional ingroup taxon (especially see Reply to comment 2.3), especially its soil species. We made a first synthesis of all the currently available information on this group with a focus on soil species traits and their role in the evolution of the soil food web. From this point of view, our synthesis is in our opinion robust, as we have argued in our point-to-point replies and in the manuscript, using the reviewers' comments as an opportunity to clarify our approaches and the validity and limitations of our results.

REVIEWERS' COMMENTS:

Reviewer #1 (Remarks to the Author):

This is a much improved submission and the authors have made some effort to address the previous concerns raised by the reviewers. I think the manuscript is now acceptable for publication in *Communications Biology*, subject to a few minor issues listed below.

INTRODUCTION

line 38: meters squared (area) or meters cubed (volume) ?

line 92: better "could have influenced" or "could have driven" ?

RESULTS

line 106: are astigmatids really that uncommon in soils. I thought that quite a lot of species live in association with insects and may effectively become part of the soil community via their host?

line 109: a taxon can only have one sister-group, so its either Prostigmata or Endeostigmata.

line 113: somewhat related to this, I'm sure that none of the endeostigmatids belong to the Parasitiformes. Do you mean that endeostigmatids belong to both Trombidiformes and Sarcoptiformes within Acariformes? Please check this against currently accepted phylogenies.

line 127: 200 mya [to be consistent with other citations of dates]

METHODS

line 304 should "burnin-in" be "burn-in" (see e.g. line 298) ?

line 306: "...selected from the literature..."

REFERENCES

line 343: Don't the italicise "E" in Early.

line 476: should "Bioninformatics" be "Bioinformatics" (see e.g. line 473).

FIGURE CAPTIONS

line 484 "angiosperms" [not Angiosperms]

line 488: "angiosperms" [not Angiosperm]

line 497: "angiosperms" [not Angiosperm]

line 505: better "supplementary information" [not informations] ?

SUPPLEMENTARY DATA

There seems to be an inconsistency with the phylogeny text and the later data in Table S1. In the first text you refer to six outgroups (1 parasitiform, 2 prostigmatans and 3 endeostigmatans), but in Table S1 the outgroups are three parasitiforms and three endeostigmatans. Please check this, correct the supplementary text accordingly, and be sure that it is also consistent with lines 112-114 of the main text where you again say there was ONE parasitiform. However, Table S1 clearly documents THREE parasitiforms: a tick, a holothyrid and an opilioacariform!

Fig. S4: here (and elsewhere) "body mass" or "bodymass". Please be consistent, and I think the former is

correct.

Table S1: "Phauloppia" [not Phalopoia]

Table S2: references "nortoni sp. nov." ? [not Nov.]

Table S2: references "County Antrim" [not Anrim]

Table S4: references why is "Tectocephus" in bold?

Jason Dunlop

Reviewer #2 (Remarks to the Author):

I have read the corrected version of the MS and I am satisfied by changes made by authors. All my concerns were discussed in details and authors explained their reasoning for accepting or rejecting it (Astigmata question). Obviously authors have mistaken in the rebuttal brief writing that one lineage of Endeostigmata belong to Parasitiformes – no, one is sarcoptiform the second trombidiform. Of course it doesn't matter for MS and I think, that the present version of it is suitable for publishing in Communications Biology.

REVIEWERS' COMMENTS:

Reviewer #1 (Remarks to the Author):

This is a much improved submission and the authors have made some effort to address the previous concerns raised by the reviewers. I think the manuscript is now acceptable for publication in Communications Biology, subject to a few minor issues listed below.

INTRODUCTION

line 38: meters squared (area) or meters cubed (volume) ?

Reply: Changed to meters squared in line 38: "... local abundance (up to 400,000 individuals m^{-2})"

line 92: better "could have influenced" or "could have driven" ?

Reply: changed to "could have influenced": "Transitions that could have influenced oribatid mite diversification or induced shifts in feeding strategies ..."

RESULTS

line 106: are astigmatids really that uncommon in soils. I thought that quite a lot of species live in association with insects and may effectively become part of the soil community via their host?

Reply: Astigmata are indeed common in soils, but are the least common of the soil mites but favor moist environments high in organic matter and naturally occur patchy or in ephemeral habitats such as decaying logs, fungal fruiting bodies, dung, carrion, sap flows, tree holes, phytotelmata, and caves (Coleman et al. 2004, in: Fundamentals of Soil Ecology; OConnor 2009, in: A Manual of Acarology by Krantz & Walter). In the soil matrix, the oribatid mites' habitat, they are indeed absent or not common.

However, we understand that the sentence in line 106 is ambiguous in its meaning and we revised this sentence, by adding more information on the specialized character of free-living astigmatid mites in lines 120-122. " This is now recognized as part of Oribatida²⁸, but strongly differs in ecology and life histories. Astigmata are mostly parasitic, and free-living species and occur patchy in moist environments high in organic matter, or in ephemeral habitats, such as decaying logs, fungal fruiting bodies, dung, carrion or tree holes, and therefore play a limited role in terrestrial soil food webs^{5,29}. "

New Reference: 29 OConnor, B. M. in A Manual of Acarology 3rd Edition (eds. Krantz, G. W. &

Walter, D. E.) Chapter 16 (Texas Tech University Press, 2009).

line 109: a taxon can only have one sister-group, so its either Prostigmata or Endeostigmata.

Reply: We agree and changed the statement about outgroups in the sentence as follows: "However, we included three Endoestigmata, which is relevant for the phylogenetic resolution among early derived oribatid mites and for fossil calibrated divergence time estimations²⁸." (now lines 124-126)

line 113: somewhat related to this, I'm sure that none of the endeostigmatids belong to the Parasitiformes. Do you mean that endeostigmatids belong to both Trombidiformes and Sarcoptiformes within Acariformes? Please check this against currently accepted phylogenies.

Reply: We agree and changed the sentence accordingly to: " We included three prostigmatid mites as a outgroup taxon to Acariformes, but ignored non-mite chelicerate taxa as outgroups in this study, because we specifically focused on trait evolution and divergence times of nodes and branches only within oribatid mites." (now lines 126-129)

line 127: 200 mya [to be consistent with other citations of dates]

Reply: We changed "my" to "million years" to avoid confusion of abbreviations, because we stated in this sentence that 25% of all branching events occurred during a time span of 200 million years (which had happened several hundred million years ago).

METHODS

line 304 should "burnin-in" be "burn-in" (see e.g. line 298) ?

Reply: Thank you very much for noticing! We changed this to the reviewer's suggestion.

line 306: "...selected from the literature..."

Reply: Changed according to the reviewer's suggestion.

REFERENCES

line 343: Don't the italicise "E" in Early.

Reply: Changed!

line 476: should "Bioninformatics" be "Bioinformatics" (see e.g. line 473).

Reply: Changed!

FIGURE CAPTIONS

line 484 "angiosperms" [not Angiosperms]

Reply: Changed throughout the manuscript!

line 488: "angiosperms" [not Angiosperm]

Reply: Changed!

line 497: "angiosperms" [not Angiosperm]

Reply: Changed!

line 505: better "supplementary information" [not informations] ?

Reply: Changed!

SUPPLEMENTARY DATA

There seems to be an inconsistency with the phylogeny text and the later data in Table S1. In the first text you refer to six outgroups (1 parasitiform, 2 prostigmatans and 3 endeostigmatans), but in Table S1 the outgroups are three parasitiforms and three endeostigmatans. Please check this, correct the supplementary text accordingly, and be sure that it is also consistent with lines 112-114 of the main text where you again say there was ONE parasitiform. However, Table S1 clearly documents THREE parasitiforms: a tick, a holothyrid and an opilioacariform!

Reply: Thank you very much for noticing! We checked alignments and trees again and found a labelling error in one of the outgroup taxa. Our dataset contains three endeostigmatid and three parasitiform mites as outgroups (please see also our reply above, to line 113). This labelling mistake does neither affect the aims of our study, nor any of the analyses, our results or their interpretation.

Fig. S4: here (and elsewhere) "body mass" or "bodymass". Please be consistent, and I think the former is correct.

Table S1: "Phauloppia" [not Phalopoia]

Reply: Changed!

Table S2: references "nortoni sp. nov." ? [not Nov.]

Reply: Changed!

Table S2: references "County Antrim" [not Anrim]

Reply: Changed!

Table S4: references why is "Tectocephus" in bold?

Reply: Changed from bold to italics!

Jason Dunlop

Reviewer #2 (Remarks to the Author):

I have read the corrected version of the MS and I am satisfied by changes made by authors. All my concerns were discussed in details and authors explained their reasoning for accepting or rejecting it (Astigmata question). Obviously authors have mistaken in the rebuttal brief writing that one lineage of Endeostigmata belong to Parasitiformes – no, one is sarcoptiform the second trombidiform. Of course it doesn't matter for MS and I think, that the present version of it is suitable for publishing in Communications Biology.

Reply: The inconsistency about outgroups and their taxonomic assignment has been revised, please see our reply above.